# Students' Scientific Evaluations of Water Resources

**Josh Medrano [1],\*, Joshua Jaffe [1], Doug Lombardi [1] [3], Margaret A. Holzer [2] and Christopher Roemmele [3]**

1. Department of Human Development & Quantitative Methodology, University of Maryland, College Park, MD 20742, USA; jjaffe13@umd.edu (J.J.); lombard1@umd.edu (D.L.)
2. Great Minds PhD Science, Washington, DC 20003, USA; missy.holzer@gmail.com
3. Department of Earth and Space Sciences, West Chester University, West Chester, PA 19383, USA; croemmele@wcupa.edu
* Correspondence: jmed7@umd.edu

**Abstract:** Socially-relevant and controversial topics, such as water issues, are subject to differences in the explanations that scientists and the public (herein, students) find plausible. Students need to be more evaluative of the validity of explanations (e.g., explanatory models) based on evidence when addressing such topics. We compared two activities where students weighed connections between lines of evidence and explanations. In one activity, students were given four evidence statements and two models (one scientific and one non-scientific alternative); in the other, students chose four out of eight evidence statements and three models (two scientific and one non-scientific). Repeated measures analysis of variance (ANOVA) showed that both activities engaged students' evaluations and differentially shifted students' plausibility judgments and knowledge. A structural equation model suggested that students' evaluation may influence post-instructional plausibility and knowledge; when students chose their lines of evidence and explanatory models, their evaluations were deeper, with stronger shifts toward a scientific stance and greater levels of post-instructional knowledge. The activities may help to develop students' critical evaluation skills, a scientific practice that is key to understanding both scientific content and science as a process. Although effect sizes were modest, the results provided critical information for the final development and testing stage of these water resource instructional activities.

**Keywords:** water resources; understanding; scientific reasoning

---

## 1. Introduction

There is an increasing recognition of the scope and complexity of water problems [1]. These challenges are global and cannot be solved only through local policies or regional policies (i.e., scope), and they are interrelated with issues of climate change and population growth, and require coordination between local, regional, and national governments (i.e., complexity). Some common water-related issues are quality (e.g., drinking and sanitation access), decline in freshwater and ecosystem conditions as a result of chemical and physical alteration, climate change shifting the locations of water supplies, and over harvesting of groundwater [2,3] and decrease in the availability of wetlands that assist in the water purification process [4]. In each of the past six years, the World Economic Forum has identified "water crises" as one of the top 5 most impactful social risks [5]. Therefore, water issues need to be better understood and addressed by students, scientists, engineers, and the public, now and in the future.

*Science, Science Education, and Scientific Evaluations*

In the past century, science education in the United States and Europe has undergone two periods of major reform [6]. The first period was at the beginning of the Cold War, when scientific productivity

was particularly valued and science curriculum geared toward developing students for science careers. The second was the 1980s, when the global economy and modernization called for the development of a more scientifically literate populace. Although national standards served to facilitate these reforms, some researchers argue that science education has not addressed the heart of science itself, which is the construction of knowledge. That is, science curricula have been focused on what needs to be known in order to do and understand science, but not how and why we know what we know [6]. More recent efforts in the area of learning sciences have been made to equally focus on epistemic and social learning goals. For instance, in 2012, one body of researchers proposed using the term "practices" instead of "skills" to emphasize that knowing the what, how, and why of science requires a coordination of ability and specific knowledge [7]. One implication of such usage points to the foundation of science, that "scientific practice is based not on rules, but on processes of perpetual evaluation and critique that support progress in explaining nature" [8].

This "purposeful evaluation and critique" is foundational to a set of scientific practices such as observations, data collection, experimentation, questioning, and constructing explanations and theories of a phenomenon. Through the practice of evaluation, students gain a deeper understanding of a concept [9–11], as well as a greater understanding of both what scientists know (i.e., scientific content) and how scientists know what they know (i.e., scientific epistemology) [12]. Such activities implementing this practice are useful when engaging in socioscientific topics—that is, topics that are controversial, socially relevant, and informed by science [13]—whether they be the causes of current climate change, impacts of fracking, or availability of freshwater resources [9–11].

In such activities, those that are evaluating may consider their own epistemic judgments. One such epistemic judgment is plausibility, which can be defined as a judgment about the potential truthfulness of an explanation [14]. Laypersons and scientists alike perform plausibility judgments that are either automatic and implicit or deliberate and explicit. Evaluations of an explanation's plausibility can lead to a shift in conceptual understanding. In a recent study, high school students' post-instructional knowledge of different scientific phenomena (i.e., causes of climate change, impacts of fracking, role of wetlands, and formation of the Earth's Moon) were found to be related to participants' evaluations, above and beyond background knowledge [11]. In another study with middle school students, these shifts persisted 6 months after instruction [9]. In sum, explicit and purposeful evaluation facilitates deep understanding when considering explanations about phenomena, especially when students consider their own epistemic judgments and compare how scientists construct and reconstruct knowledge.

Despite the evidence, there is a gap in what laypersons and scientists might find plausible (i.e., a plausibility gap; [9]). Take the role of wetlands, for instance. One person might view wetlands as protection, recharging groundwater, and maintenance of valuable habitat for fish, birds and other animals, and plants [4]. Another person might view wetlands as having little social and environmental benefit because they flood during heavy rainfall, provide a hotbed for mosquitoes and other pests, and prohibit development of commercial and residential areas. We describe these viewpoints as "explanatory models" because they can provide mental and conceptual representations that can be used when solving a problem or analyzing a situation [15]. Our project, and specifically the present study, comparatively examines the effectiveness of instructional scaffolds in authentic secondary classroom settings, with the goal of gauging how Earth and/or environmental science students can deepen their knowledge about water resources through scientific evaluations and judgments.

## 2. Background and Theoretical Framework

The present study built upon the characterization of scientific thinking as the "consciously controlled evaluation of [explanations] in the light of [evidence]" [16]. Explanations may refer to accounts of how phenomena unfold that may lead "to a feeling of understanding in the reader/hearer" [17]; may include fully developed theories and models, as well as facets of theories and models containing essential kernels of theory [18,19]; and may refer to "explicit applications of

theory to a specific situation or phenomenon, perhaps with the intermediary of a theory-based model for the system under study" [7]. A scientific explanation may be valid from its plausibility, especially the plausibility of the explanation relative to alternatives [20]. Students and the general public engage in evaluations based on the plausibility of explanations, often implicitly (i.e., without much or any thought; [14]). The present study uses a theoretical model that views evaluation as a central component in the dynamic process of students' appraisal and/or reappraisal of an explanation's plausibility [14]. This model also posits that evaluations and plausibility reappraisal may facilitate deeper knowledge of science. As a foundation for the present study, the following subsections provide more details on the theoretical connections of evaluation, plausibility judgments, and knowledge.

## 2.1. Evaluation

Evaluation is central to scientific practice [7,8] and is iterative in the scientific process. Scientists and engineers evaluate, for example, whether a model fits a set of collected data, and conclude whether a readjustment of the model is necessary. Researchers evaluate whether their design or data analysis methods conform to a theory or are valid in testing a theory and may conclude that more experiments are needed. When scientists engage in evaluation, they are asking whether the data fits their model, as well as what the strengths and limitations are in their theories, designs, and explanations. Researchers can also compare research designs and which ones are a better design for the theory. Recent science education reform efforts place this process of evaluation at the core of scientific activities that students should engage in during science instruction [7]. Ford [8] specifically suggested that a process of critique and evaluation (i.e., critical evaluation) is fundamental to each of the classroom scientific practices embedded within the *Next Generation Science Standards* [21]. For example, science students may evaluate whether their data answers a theory, or which of two hypotheses the data fits more. Previous research has shown that students' evaluations of connections between evidence and alternative models differ qualitatively and that these differences are predictive of post-instructional knowledge [10]. Qualitative differences specifically reflect the scientific accuracy of students' evaluation and the reasoning quality of their explanations (see Section 3.3 and Appendix D for the levels of explanations). Therefore, helping students to become more critically evaluative as they learn about science will potentially lead them to be more scientifically literate. One way to promote evaluative processes is to explicitly engage students in judgments about knowledge and knowing (i.e., epistemic judgments, such as judging the plausibility of scientific explanations) [14].

## 2.2. Plausibility Judgment

Plausibility is a tentative epistemic judgment that facilitates the construction and reconstruction of knowledge both in science and in science classrooms. Researchers have implicated plausibility judgments in enabling conceptual change [22,23], as well as in facilitating co-construction of knowledge in discourse associated with collaborative argumentation [6,24,25]. Lombardi and colleagues [14] developed a theoretical account of how explicit and critical evaluations about novel explanations may influence appraisals and reappraisals of plausibility, which in turn might influence knowledge reconstruction. In their theoretical framework, the meaning of plausibility was discussed from both the philosophy of science and science studies perspectives; specifically, how scientists incorporate plausibility judgments into their evaluations of explanations and how plausibility as an epistemic judgment of potential truthfulness is an outcome of individuals' evaluations (see, e.g., [17]). Studies, including the present one, have sought to test this theoretical model, with the aim to find ecologically valid instructional activities that could promote critical evaluation, plausibility reappraisal, and knowledge shifts about scientific topics.

## 2.3. Instruction that Promotes Evaluation and Plausibility Reappraisal

The theoretical relationship between evaluation and plausibility judgments has been tested as part of a multi-year project that has focused on finding instructional treatments that can activate

students' critical evaluations. In this project, we employ an instructional treatment known as the model–evidence link (MEL) diagram, where students evaluate connections between lines of scientific evidence and two alternative and competing explanations of a phenomenon (e.g., a scientific and a non-scientific alternative explanation about the causes of current climate change). In one study using the MEL, high school students' plausibility judgments shifted toward scientifically accepted explanations and increased their overall knowledge about the topics (i.e., climate change, fracking, wetlands, and the formation of the Earth's moon) [11]. Furthermore, structural equation modeling showed that plausibility reappraisal partially mediated a positive relationship between evaluation and post-instructional knowledge. Effect sizes varied depending on topic and instructional context, such as that a large effect size existed for some topics (e.g., connections between fracking and earthquakes) and a small effect size for others (e.g., the importance of wetlands on ecosystem services). Thus the authors concluded that the MEL motivated students to cognitively engage in practices that helped them think more scientifically.

A second study [26] confirmed this finding, while also comparing the MEL, where students evaluate connections diagrammatically, to two other treatments: Mono-MEL, where students evaluate connections between lines of evidence and only one explanation (i.e., the scientific alternative), and the model-evidence link table (MET), where students evaluate connections between lines of evidence and two models using tables and letter codes. Fine-grained comparative analyses of plausibility scores showed that at post-instruction, but not pre-instruction, MEL plausibility scores were significantly greater than both Mono-MEL and MET, and that there was a significant shift in plausibility scores in both MEL and MET, but not Mono-MEL, with larger effect size for MEL than for MET. Analysis of knowledge scores showed no differences in pre-instruction and post-instruction, but significant shifts in MEL and MET, but not in Mono-MEL. Overall, the fine-grained analyses showed that considering two models is more effective at shifting plausibility judgments and knowledge than considering only one model. A secondary analysis considering all variables together using structural equation modeling (SEM) indicated that MEL had a greater influence on evaluations than either the MET or Mono-MEL. Thus, the two studies above showed that the treatments engaged students' evaluation when considering alternative explanations about Earth and space science phenomena and that their levels of evaluations are significantly related to plausibility appraisals and knowledge of the phenomena [11,26].

### 2.4. Current Study

This present study examined and compared how plausibility judgements and knowledge construction shifted based on the two different types of MEL diagrams: pre-constructed model–evidence link diagrams (pcMEL), and a build-a-MEL (baMEL) diagram. (Figure 1) The pcMEL diagrams present four lines of scientific evidence and two models (scientific alternative and other alternative), as in previous studies [11,25]. In the baMEL, students constructed their own diagram to extend the topic and selected four new evidence lines from eight choices and two new alternative explanatory models from three choices (scientific, engineering, and non-scientific alternatives). With the completion of the MEL diagrams, students will then select and further analyze model–evidence relationships and evaluate the relationship between the two. By building and evaluating their own MEL diagrams, we hypothesized that students would activate greater levels of conceptual agency (i.e., as authors of their knowledge contributions, students would have increased accountability to the classroom community and explicit authority to reason) [27]. If so, then a theoretical and practical contribution of the present study could suggest that deeper conceptual agency activated by the baMEL instructional scaffold has the potential for: (a) strengthening students' evaluations about the connections between lines of scientific evidence and alternative explanatory models; (b) promoting greater shifts in plausibility toward the scientific alternative; and (c) increasing science learning about water resources. This is in line with a theoretical account of increased transfer effects associated with conceptual agency [27]. In the present study, the MEL diagrams focused on two topics concerning water resources, specifically wetlands' role in ecosystems (the pcMEL; Figure 1a) and future freshwater availability (the baMEL; Figure 1b).

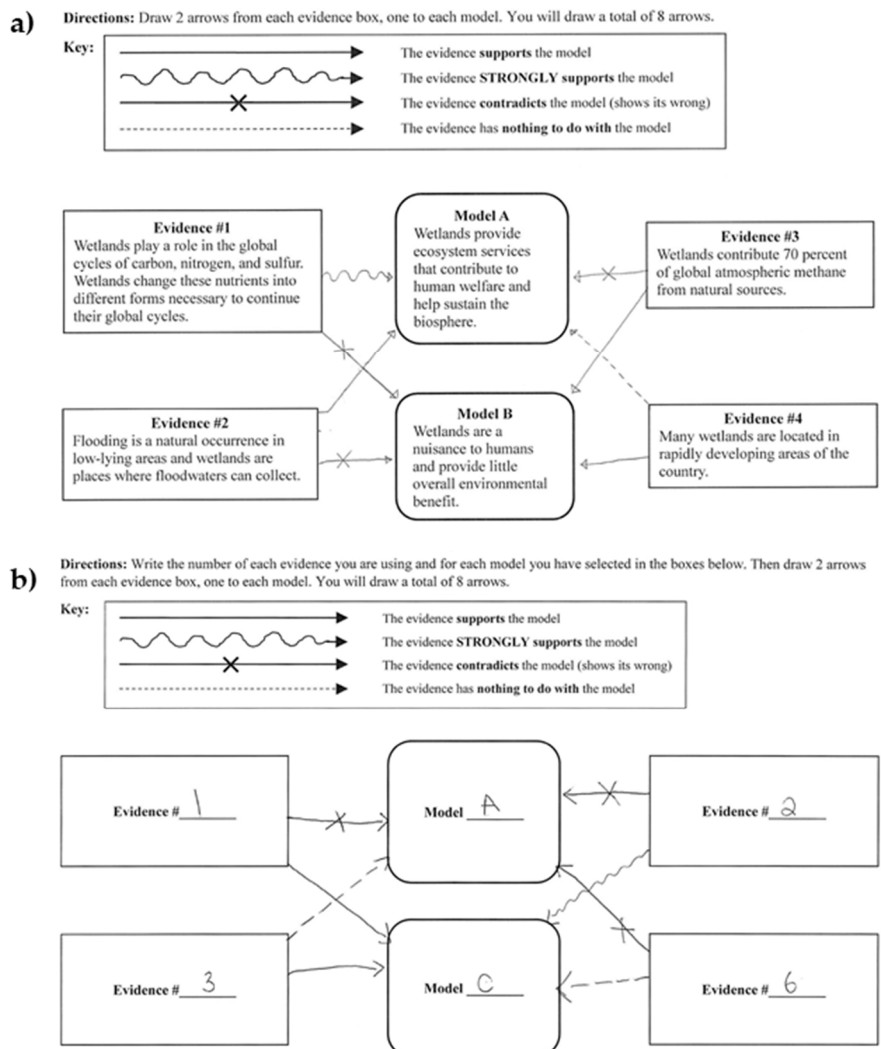

**Figure 1.** A student example of the wetland pre-constructed model–evidence link (pcMEL) (**a**) and the freshwater build-a-MEL (baMEL) baMEL (**b**) diagrams. Refer to Appendix A for more details on the lines of evidence and models in each instructional treatment.

The topics of wetlands' role in ecosystems and freshwater availability are two important socio-scientific topics found in many Earth and/or environmental science curricula [28]. These topics were part of the curricular scope and sequence in middle school and high school classes participating in year two and three of a multiyear project. The primary goal was to test the impact that both MEL diagrams have on: (a) promoting students' evaluations when gauging the connections between lines of scientific evidence models and alternative explanations; (b) promoting plausibility appraisals toward a more scientific stance; and (c) deepening students' knowledge of the phenomenon (e.g., water resources. Exposure to scientific knowledge is, in and of itself, insufficient for engaging students in evaluative and scientific thinking [29]. However, instruction that scaffolds evaluative thinking, such as the pcMEL and baMEL activities, may result in stronger engagement where students think more scientifically [11].

In the present study, the wetlands topic was presented to students in the pcMEL platform with a scientific and non-scientific alternative. Freshwater availability was presented on a baMEL diagram with a scientific, engineering and non-scientific alternative. The present study examined students' evaluations and learning about the connections between lines of scientific evidence and alternative explanations of important water resources topics that are pertinent in today's society; we examined the following research questions:

1. What are the levels of students' evaluations when they engage in two instructional treatments (i.e., the pcMEL and baMEL) and how do students' plausibility judgements and knowledge change over the course of these two instructional treatments?

2. How do relations between students' evaluations, plausibility judgements, and knowledge compare between the pcMEL and baMEL?

## 3. Materials and Methods

### 3.1. Participants

Participants were middle school (Grades 5–8; Teachers 05 and 06) and high school (Grades 9–12; Teachers 01 and 58) students, from two school districts in the United States. Both middle school classrooms and one high school classroom were in a medium-sized suburban district in the Mid-Atlantic US, and one high school teacher was located in a suburban district in the Southeast US. Data were collected from 14 class periods, with a total of 248 students. However, only 108 students completed all activities and were thus included in the final analysis. Participants' race and ethnicity reflected the overall compositions of their individual schools (Appendix A).

### 3.2. Instructional Treatments

In this study, we implemented MEL diagrams covering two different water resources topics: wetlands' role in ecosystems (i.e., Wetlands pcMEL) and availability of freshwater resources (i.e., Freshwater baMEL). These topics included lines of scientific evidence and explanatory models about water-related phenomena (see Appendix A) that students could evaluate. Both topics were part of the curricular scope and sequence in the classrooms that participated in the present study.

3.2.1. Wetlands pcMEL

Wetlands' role in ecosystem services presents an important and compelling socio-scientific topic. There has also been heavy debate on the cost and benefits of either preserving wetlands or using the land for alternative uses [30]. Two explanatory models were selected for this pcMEL topic, the scientific explanation that wetlands provide ecosystem services that contribute to human welfare and help sustain the biosphere, and the alternative explanation, wetlands are a nuisance to humans and provide little overall environmental benefits. The accompanying lines of evidence discussed the role wetlands play in global cycles of carbon, nitrogen, and sulfur; the effect that wetlands have on flooding in low-lying areas; the contribution of global atmospheric methane from wetlands; and the location of wetlands compared to developing areas. Students specifically evaluated relevant scientific information to make judgments on the relation between the evidence and the model of explanation (for more detail see Appendix C). We acknowledge that individuals may look at other types of evidence (e.g., economic benefits and costs) when evaluating such explanations about wetlands, but for the purpose of comparison, we kept the pcMEL focused on scientific lines of evidence.

The first section of the MEL activity introduced the students to the four evidence texts and two explanatory models, the scientific and the non-scientific alternatives. On the Wetlands pcMEL diagram (Figure 1a), the explanatory models are in the center of the page in two separate boxes (explanatory texts had no labeling indicating which one is scientific and alternative explanation) with short summaries of the four evidence texts surrounding the models, each of which had a corresponding "evidence text" that is about a page in length. The detailed versions of the evidence texts also included graphs and diagrams to further explain the evidence. Participants were instructed to draw different types of arrows from each evidence text to both models based on how well they thought the evidence supports the model. Four different types of arrows were used; a squiggly arrow indicated the participant believes that the evidence strongly supports the model, a straight arrow indicated that the evidence supports the model, a dotted line arrow indicated the evidence had nothing to do the model,

and a line with an "X" in the middle of it indicated that the evidence contradicts the model. Overall, the participants drew eight arrows in total (Figure 1a).

### 3.2.2. Freshwater baMEL

Freshwater availability is an especially pertinent issue today with the world's population increasing and the current climate change crisis. However, unlike the climate change crisis, which is a central issue in the media and the subject of strong beliefs in individuals (whether they are scientific or a non-scientific alternative belief), the freshwater availability crisis has not gained the same popularity [31]. The freshwater topic was presented to middle school and high school students using the baMEL activity. We designed three plausible explanatory models about the freshwater availability situation: the scientific explanation that Earth has a shortage of freshwater, which will worsen as our world's population increases [32]; the engineering alternative where engineered solutions can help mend the global shortage of freshwater; and the non-scientific alternative that freshwater is abundant regardless of global climate and population situation [33]. The eight lines of evidence of freshwater availability covered subtopics of freshwater resources, reclamation costs, underground freshwater, glacier ice mass, temperature change, drought, and microclimates (see Appendix C).

The first section of the freshwater availability MEL activity introduced the students to the eight lines of evidence and the three explanatory models: the scientific, engineering, and non-scientific alternative. Unlike the Wetlands pcMEL diagram, where models were provided, the Freshwater baMEL diagram contained two blank boxes in the middle in which participants wrote the letter of the explanatory models they selected (A, B, and/or C), and four blank boxes around the edge where participants wrote the number of the lines of evidence they selected. In other words, participants filled in these blanks by selecting two of the three explanatory models and four of the eight lines of evidence. Compared to the Wetlands pcMEL, the Freshwater baMEL gave participants the opportunity to pick which evidence text and exploratory models they would like to incorporate into their MEL diagram, thus increasing the potential for conceptual agency [27]. Similar to the Wetlands pcMEL, participants connected the evidence texts to models using the four different types of arrows, and drew eight total arrows (Figure 1b).

### 3.3. Evaluation/Explanation Task

The second section of the MEL activity is referred to as the "Explanation Task" (ET; Figure 2). Participants picked either one or two of the connections that they drew from the first MEL activity and gave their explanation on why they drew a particular type of arrow (i.e., an explanation about the evaluation of the strength between a particular line of evidence and a particular model). Using a scoring system and rubric developed by Lombardi and colleagues [9], coders rated explanations for different levels of evaluation using a rubric (see Appendix D for description and examples): 1 = Erroneous, 2 = Descriptive, 3 = Relational, 4 = Critical. The categories established well-defined levels of evaluation to represent the accuracy and elaboration present in participants' responses. To establish coding reliability, 25% of the ETs were scored by two coders (the first and second authors), with an intraclass correlation coefficient (ICC) of 0.72, indicating a high reliability between the coders and full consensus reached after consultation between the two coders. One coder completed scoring for the remainder of the ETs.

### 3.4. Plausibility Judgment

For both the Wetlands pcMEL and Freshwater baMEL, students were instructed to rate the plausibility of all explanatory models, both pre- and post-instruction. For the Freshwater baMEL, students recorded their plausibility judgments for all three explanatory models, while for the Wetlands pcMEL, students recorded their plausibility judgments for the two explanatory models. Students gauged the plausibility of each model using a 1–10 scale (1 = greatly implausible and 10 = highly plausible), based on previous measures used by Lombardi and colleagues [11,28]. Because the Wetlands pcMEL offered only two explanatory models, scores were calculated as the rating of the scientific model minus the

alternative. The Freshwater baMEL offered three different explanatory models (scientific, engineering, and non-scientific alternatives), and therefore, three different scores were calculated: scientific minus non-scientific, scientific minus engineering, and engineering minus non-scientific. Scores could range on a scale from −9 to +9, where positive scores indicated that participants judged the scientific (or engineering) model as more plausible than the alternative model, with negative scores indicating participants judged the non-scientific as being more plausible than the scientific (or engineering).

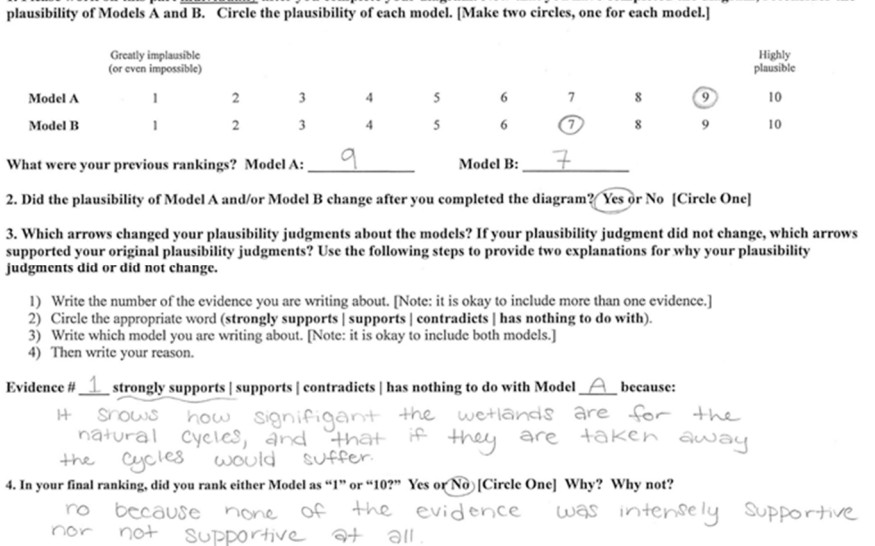

**Figure 2.** Plausibility Ratings and Explanation Tasks for pcMEL. A similar form was used for baMEL.

### 3.5. Knowledge

For both the Wetlands pcMEL and Freshwater baMEL, participants completed a multi-item knowledge survey instrument (at pre- and post-instruction; see Appendix E). The freshwater knowledge survey contained twelve items and the wetlands knowledge survey contained five items. Students ranked each item on a 5-point Likert scale (1 = strongly disagree and 5 = strongly agree) on their knowledge of how scientists would agree with each item statement. At least one question in each set addressed each line of scientific evidence. Questions were constructed in two different formats; some statements were negatively worded (i.e., in effect scientists would disagree with these knowledge statements) and we reverse coded these statements. McDonald's omega coefficients (ω) indicated that the knowledge scale for each scaffold and each time points were sufficiently reliable for a pilot study (Wetlands pcMEL pre: ω = 0.42, post: ω = 0.56; Freshwater baMEL pre: ω = 0.76, post: ω = 0.69). Overall knowledge scores represented the average of the ratings for each survey. See Appendix E for a list of all items that we used to survey understanding of the wetlands and freshwater topics.

### 3.6. Procedures

During the summer prior to the present study, classroom teachers participated in a three-day professional development workshop with the project team. The workshops focused on introducing and practicing using the pcMEL and baMEL activities, going over the content and pedagogical strategies for effective classroom implementation, and planning for the upcoming year's implementation of MEL activities. To maintain some uniformity in instruction, the teachers agreed to introduce each activity at the beginning of a unit prior to any instruction about the topic, with the Wetlands pcMEL activities taught first and the Freshwater baMEL activities taught second. The teachers agreed to follow the lesson plans, as specified at the workshop and as found in the teacher guides. In other words, the teachers presented the activities using the instructions present on the student materials. Students completed all of the activities over the course of an instructional unit focused on water resources.

Near the beginning of the school year and prior to participating in either treatment, students performed the "plausibility ranking task" as an introduction to the ideas of plausibility and critical evaluation. This task asked students to rank the importance of different types of evidence for determining the plausibility of an explanatory model. These four types of evidence were the same as the links that students later indicated on the activities: evidence that supports the model, strongly supports it, contradicts it, or has nothing to do with it. After ranking the importance of each from 1–4, they read a small passage on falsifiability that states scientific ideas cannot be proven but are rather disproven through opposing evidence, and were then asked to rank the types of evidence again. This provided an introduction to the idea of plausibility for students and an initial look at the comfort with which they can evaluate the roles of scientific evidence.

For a given treatment, students began by completing the knowledge survey and model plausibility ratings for each explanatory model on that topic (see Figure 3). At this time, teachers also engaged the class in an unscripted short discussion of the model(s) and the idea of plausibility, to clarify misunderstandings and address general questions about the topic. When completing the Wetlands pcMEL activity, students read the evidence texts and completed the diagram in small groups. Next, they worked individually to write up the explanation task. The activity ended with the second iteration of the model plausibility ratings and wetlands knowledge survey. When completing the Freshwater baMEL activity, students first read the texts for all eight lines of evidence, and were introduced to the three alternative models explaining the phenomenon. Then small groups of students worked together to select four lines of evidence from the eight available and two alternative models from the three available. The students used these four lines of evidence and two models to construct a MEL diagram, which they then completed (i.e., by drawing arrows). Similar to the Wetlands pcMEL activity, students then worked individually to complete the explanation task. The activity ended with the second iteration of the model plausibility ratings and freshwater knowledge survey. Upon completion of this sequence, teachers moved on to teaching their regular instructional unit. Each MEL activity took place over about two regular class periods (~90 min total), with no appreciable difference in instructional time between the Wetlands pcMEL and Freshwater baMEL.

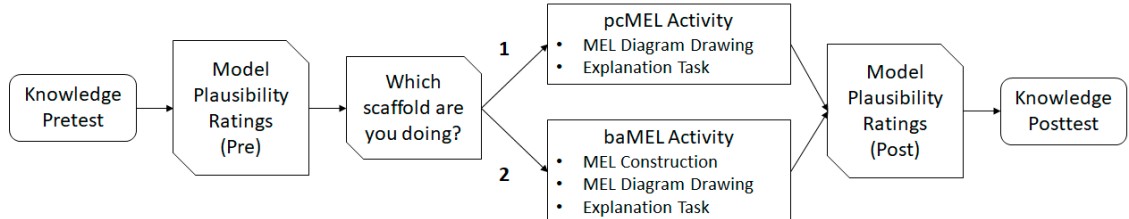

**Figure 3.** Students first completed the knowledge survey and the plausibility ratings. Then, students completed each pcMEL (1) and baMEL (2) activity over about two regular periods, after which they completed another knowledge survey and plausibility ratings. The only difference between the two MEL activities is that in the baMEL, students were chosen to pick the models and evidence statements.

## 4. Results

We present the results in two sections. The first section addresses Research Question 1: What were the levels of students' evaluations when they participated in the two instructional treatments (i.e., the pcMEL and baMEL) and how did students' plausibility judgements and knowledge change over the course of these two instructional treatments? This section represents a fine-grained comparative analysis of the effectiveness of the instructional treatments (Wetlands pcMEL and Freshwater baMEL). The second section addresses Research Question 2: How did relations between students' evaluations, plausibility judgements, and knowledge compare between the pcMEL and baMEL? This second section represents a much broader survey of the overall relations between participants' evaluations, plausibility judgments, and knowledge.

Prior to presenting these results, we discuss the data screening that we first conducted on the data. We specifically screened the data for outliers and to ascertain alignment with assumptions inherent in ordinary least-squares analyses about the normality and linearity of the sample, as well as assumptions about the equality of homogeneity of variance-covariance matrices. We found no univariate outliers. We also concluded that the assumptions of normality and homogeneity of variances were met, after examining skewness and kurtosis, normal probability plots, and conducting Levene's tests.

*4.1. Research Question 1*

In the following analyses, we report effect sizes to gauge the robustness (i.e., strength) of differences. We specifically report generalized effect sizes ($\eta^2_G$) for the analyses of variances (ANOVAs) we conducted, with 0.02, 0.13, and 0.26 as cutoffs or approximate values of small, medium, and large effect sizes [34].

### 4.1.1. Evaluation

We ran a mixed-model univariate ANOVA with evaluation scores as the dependent variable, instructional treatment as a within-subjects factor, and whether students were in high school or middle school (school level) as a between-subjects factor. The ANOVA revealed that there was no interaction effect between treatment and school level, $F(1,106) = 0.02$, $p = 0.89$. However, the main effects were significant for both instructional treatment, $F(1,106) = 7.40$, $p < 0.05$, $\eta^2_G = 0.03$ (small effect size) and students' school level, $F(1,106) = 5.33$, $p < 0.05$, $\eta^2_G = 0.03$ (small effect). As shown in Figure 4, students' evaluation scores were higher for Freshwater baMEL ($M = 2.19$, $SD = 0.88$) than the Wetlands pcMEL ($M = 1.91$, $SD = 0.80$), and that middle school students' evaluation scores ($M = 2.18$, $SD = 0.84$) were higher than high school students' ($M = 1.90$; $SD = 0.84$).

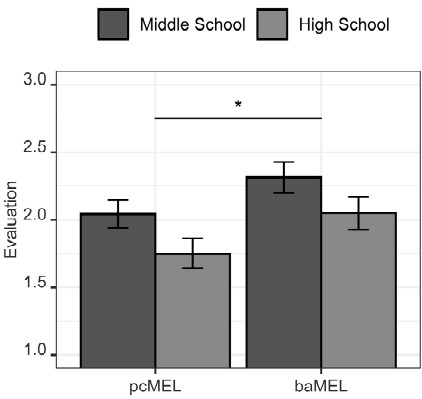

**Figure 4.** Evaluation scores (range: 1 (erroneous description)–4(critical evaluation)) for each instructional treatment. Errors bars indicated ±1 standard error. Asterisk indicates a statistically significant difference between treatments, * $p < 0.05$.

### 4.1.2. Plausibility

A mixed-model univariate ANOVA was run with Wetlands pcMEL plausibility scores as the dependent variable, time (pre- and post-instruction) as the within-subjects factor, and school level as the between-subjects factor. The ANOVA revealed no interaction effect, $F(1,106) = 0.07$, $p = 0.79$. However, the ANOVA revealed that the main effect was significant for school level, $F(1,106) = 8.73$, $p < 0.05$, $\eta^2_G = 0.06$ (small to medium effect size), but not for time, $F(1,106) = 0.37$, $p = 0.55$. As shown in Figure 4, high school participants' pcMEL plausibility scores ($M = 3.01$, $SD = 2.58$) were greater than middle school participants' scores ($M = 1.40$; $SD = 3.47$).

Prior to running the analyses for the Freshwater baMEL, we calculated two plausibility scores (C-A, the difference between the scientific and non-scientific alternatives; and B-A, the difference between the engineering and non-scientific alternatives). Higher values indicate that the student viewed the scientific or engineering model as more plausible than the nonscientific model.

We ran a repeated measures ANOVA with C-A plausibility scores as the dependent variable, time as the within-subjects factor, and school level as the between-subjects factor. The ANOVA revealed no interaction effect between time and school level, $F(1,106) = 1.03$, $p = 0.31$. However, there was a significant main effect for time, $F(1,106) = 5.48$, $p < 0.05$, $\eta^2_G = 0.01$ (small effect size), but not for school level, $F(1,106) = 2.99$, $p = 0.09$. As shown in Figure 5, students increased their plausibility scores from pre- ($M = 3.55$, $SD = 2.96$) to post-instruction ($M = 4.28$, $SD = 2.98$). We also ran a repeated measures ANOVA with B-A plausibility scores as the dependent variable, time as the within-subjects factor, and school level as the between-subjects factor. The ANOVA revealed no interaction effect between time and school level, $F(1,106) = 0.02$, $p = 0.89$. However, the main effect was significant for time, $F(1,106) = 9.55$, $p < 0.01$, $\eta^2_G = 0.03$ (small effect size), but not for school level, $F(1,106) = 0.01$, $p = 0.93$. As shown in Figure 5, students increased their plausibility scores from pre- ($M = 2.05$, $SD = 2.88$) to post-instruction ($M = 3.06$, $SD = 3.06$).

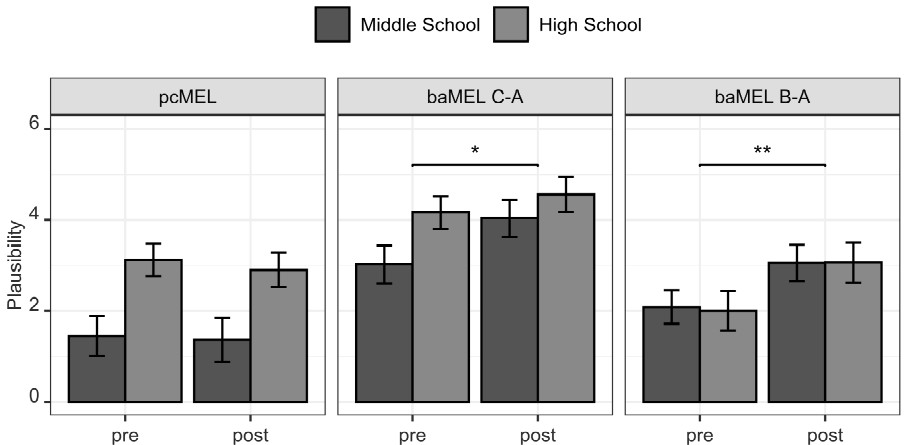

**Figure 5.** Plausibility scores (range: 1 (highly implausible)–9 (highly plausible)) for each instructional treatment. Errors bars indicated ±1 standard error. Asterisks indicate a statistically significant difference between pre- and post-instruction, * $p < 0.05$, ** $p < 0.01$.

### 4.1.3. Knowledge

A three-way mixed ANOVA was run with instructional treatment and time as within-subjects factors and school level as a between-subjects factor. There were no significant interaction effects between the three variables, $F(1,106) = 0.67$, $p = 0.41$, and between any two of the variables, all $F(1,106)$ values $< 0.67$, $p$ values $> 0.42$. However, the main effects were significant for both students' school level, $F(1,106) = 15.16$, $p < 0.001$, $\eta^2_G = 0.07$ (small to medium effect size) and time, $F(1,106) = 4.02$, $p < 0.05$, $\eta^2_G = 0.004$ (small effect size). As shown in Figure 6, knowledge scores were greater for high school participants ($M = 3.69$, $SD = 0.45$) than for middle school participants ($M = 3.43$, $SD = 0.49$). Knowledge scores also increased from pre- ($M = 3.52$, $SD = 0.47$) to post-instruction ($M = 3.58$, $SD = 0.51$). Follow-up pairwise comparisons showed that the only significant changes in knowledge scores were an increase for the Freshwater baMEL scores from pre- ($M = 3.53$, $SD = 0.45$) to post-instruction ($M = 3.61$, $SD = 0.43$), $t(107) = 2.39$, $p < 0.05$, and an increase in Freshwater baMEL scores for middle school participants from pre- ($M = 3.41$, $SD = 0.44$) to post-instruction ($M = 3.52$, $SD = 0.45$), $t(57) = 2.37$, $p < 0.05$, Cohen's $d = 0.25$ (small to medium effect size).

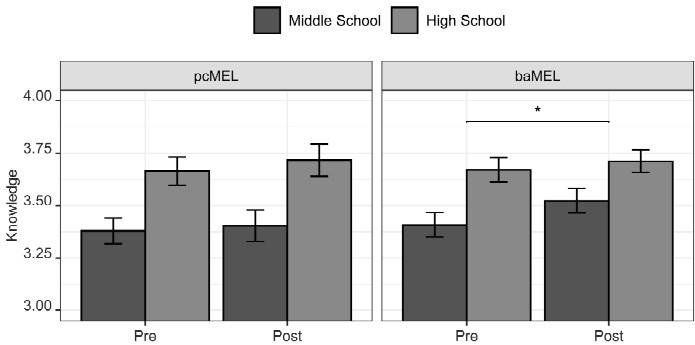

**Figure 6.** Knowledge scores (range 1 (strongly disagree with scientific)–5 (strongly agree with scientific)) for each instructional treatment. Errors bars indicated ±1 standard error. Asterisk indicates a statistically significant difference between pre- and post-instruction for middle school students completing the Freshwater baMEL, * $p < 0.05$.

### 4.2. Research Question 2

We conducted a structural equation modeling (SEM) analysis to examine relations between evaluation, plausibility (pre- and post-instruction) and knowledge (pre- and post-instruction) scores, for both topics (wetlands and freshwater resources). We specifically conducted partial least squares (PLS) analyses using Warp PLS v.4.0 statistical software [35], to examine the various relationships between the different variables. The partial least-squares method is based on ranked data and is distribution-free, which allows for more statistical power without compromising or inflating the chance for Type I errors for a large range of sample sizes and variation of group sizes [36]. PLS-SEM analyses have been used increasingly often in social science research [37] and are being used more frequently in educational research [11,38].

We constructed the latent variables in our two models (evaluation; pre- and post- instructional plausibility; and pre- and post-instructional knowledge) using scores for the two topics. For the Wetlands pcMEL SEM, latent variable scores were equivalent to the single variable scores that we used in the fine-grained analysis to address Research Question 1. We also did this for the Freshwater baMEL SEM, except the plausibility latent variable was constructed using both the C-A and B-A scores. Tables 1 and 2 show the bivariate correlations, means, and standard deviations for each variable in both the Wetlands pcMEL and Freshwater baMEL SEMs.

To check SEM validity and fit, we used multiple indices. Goodness-of-fit (GoF) examined the overall model prediction based on how well the measurements fit the model [39]. A model has a large explanatory power when GoF is greater than 0.36, with unacceptable explanatory power when GoF is less than 0.1 [40]. Average part coefficient (APC) and average coefficient of determination across the model (average $R^2$ or ARS) should give the model different relationship values that are significantly different from zero (assuming $p < 0.05$ as criteria). The average variance inflation factor (AVIF) for model parameters and average full collinearity VIF (AFVIF) were used to analyze the degree of collinearity between the models. AVIF and AFVIF values higher than 3.3 indicate redundancy between variables in the model [41]. GoF, APC, ARS, AVIF, and AFVIF were analyzed in both MEL topics. Table 3 shows validity and fit indices for the two SEMs, which show that each model had a strong effect size indicated by appreciable GoF values and ARS, and were valid based on other indices.

**Table 1.** Descriptive Statistics and Correlations between the Wetlands pcMEL structural equation modeling (SEM) latent variables.

|  | *M* | *SD* | 1 | 2 | 3 | 4 |
|---|---|---|---|---|---|---|
| 1. Evaluation | 1.91 | 0.80 | | | | |
| 2. Knowledge-Pre | 3.51 | 0.49 | 0.12 | | | |
| 3. Knowledge-Post | 3.55 | 0.58 | 0.09 | 0.56 ** | | |
| 4. Plausibility-Pre | 2.22 | 3.04 | 0.11 | 0.27 ** | 0.18 | |
| 5. Plausibility-Post | 2.07 | 3.34 | 0.25 ** | 0.22 ** | 0.28 ** | 0.67 ** |

** indicates $p < 0.01$.

**Table 2.** Descriptive Statistics and Correlations between the Freshwater baMEL SEM latent variables.

|  | *M* | *SD* | 1 | 2 | 3 | 4 | 5 | 6 |
|---|---|---|---|---|---|---|---|---|
| 1. Evaluation | 2.19 | 0.88 | | | | | | |
| 2. Knowledge-Pre | 3.53 | 0.45 | −0.18 | | | | | |
| 3. Knowledge-Post | 3.61 | 0.43 | −0.02 | 0.68 ** | | | | |
| 4. Plausibility-C-A-Pre | 3.55 | 2.96 | −0.02 | 0.27 ** | 0.35 ** | | | |
| 5. Plausibility-C-A-Post | 4.28 | 2.98 | 0.01 | 0.22 ** | 0.39 ** | 0.44 ** | | |
| 6. Plausibility-B-A-Pre | 2.05 | 2.88 | −0.01 | 0.07 | 0.10 | 0.64 ** | 0.26 ** | |
| 7. Plausibility-B-A-Post | 3.06 | 3.06 | 0.07 | 0.20 * | 0.29 ** | 0.39 ** | 0.67 ** | 0.35 ** |

* indicates $p < 0.05$. ** indicates $p < 0.01$. C-A indicates the difference in between the scientific and non-scientific alternatives. B-A indicates the difference between the engineering and non-scientific alternatives.

**Table 3.** Validity and fit indices for the pcMEL and baMEL SEMs. GoF, goodness-of-fit; APC, average part coefficient; ARS, average $R^2$; AVIF, average variance inflation factor; AFVIF, average full collinearity VIF.

|  | *pcMEL* | *baMEL* |
|---|---|---|
| 1. GoF | 0.545 | 0.42 |
| 2. APC | 0.261 | 0.239 |
| 3. ARS | 0.297 | 0.232 |
| 4. AVIF | 1.050 | 1.041 |
| 5. AFVIF | 1.615 | 1.484 |

Figure 7 shows the two SEMs. The arrangement of variables and pathway connection structure is based on Lombardi and colleagues' [14] theoretical model relating to evaluation, plausibility, and knowledge. Standardized path values shown on the figures indicate relative strength of the seven model pathways. We also compare the two SEMs in Figure 8, which shows a scatter plot of individual pathway effect sizes, which is another indicator of the strength of individual pathways. First, the models shown in Figure 8 verify Lombardi and colleagues' [14] theory that the indirect pathways between evaluation and post-instructional knowledge (i.e., the two pathways from evaluation to post-instructional plausibility to post-instructional knowledge) were stronger than the direct pathway between evaluation and post-instructional knowledge. This was the case for both the Wetlands pcMEL and the Freshwater baMEL. Second, in comparing the two models (Figure 8), although the strength of the pathway between evaluation and post-instructional plausibility was roughly the same for the two MELs, Pathway #7 between post-instructional plausibility and post-instructional knowledge was appreciably stronger (~3.5 times stronger) for the Freshwater baMEL. Further, the strength of Pathway #2 between plausibility pre-instruction and plausibility post-instruction in the Freshwater baMEL was only a fraction (~2%) of the Wetlands pcMEL. Taken together, these two comparative results reveal that the Freshwater baMEL deactivated prior plausibility judgments more effectively than the Wetlands pcMEL, with students relying more on their evidence to model link evaluations facilitated by the Freshwater baMEL.

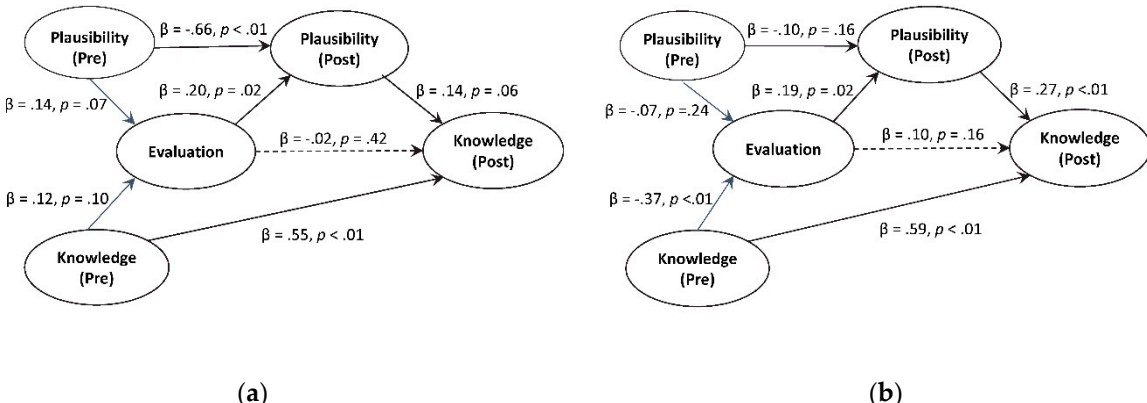

(**a**)                                                                                               (**b**)

**Figure 7.** Comparison of pathway effect sizes between the Wetlands pcMEL (**a**) and the Freshwater baMEL (**b**). Pathway effect size values are unbiased indices that are compatible with partial least squares (PLS)-SEM and essentially equivalent to Cohen's $f^2$ coefficient [34], with small effect size = 0.02, medium effect size = 0.15, and large effect size = 0.35. The dashed line indicates when effect sizes are equivalent in the two MELs. Points above and to the left of the dashed line indicated when effects sizes are greater for the Freshwater baMEL. Points below and to the right of the dashed line indicate when effect sizes are greater for the Wetlands pcMEL.

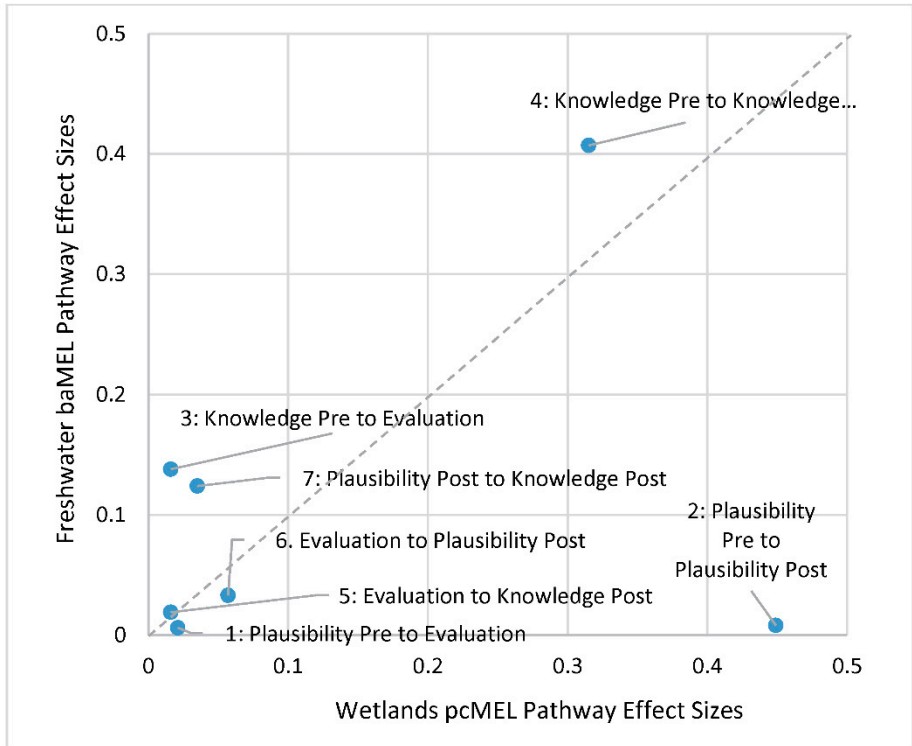

**Figure 8.** Comparison of pathway effect sizes between the Wetlands pcMEL and the Freshwater baMEL. Pathway effect size values are unbiased indices that are compatible with PLS-SEM and essentially equivalent to Cohen's $f^2$ coefficient [34], with small effect size = 0.02, medium effect size = 0.15, and large effect size = 0.35. The dashed line indicates when effect sizes are equivalent in the two MELs. Points above and to the left of the dashed line indicated when effects sizes are greater for the Freshwater baMEL. Points below and to the right of the dashed line indicate when effect sizes are greater for the Wetlands pcMEL.

### 4.3. Results Summary

The ANOVA results suggest that Freshwater baMEL had a modest advantage in effectiveness compared to the Wetlands pcMEL. Specifically, practical interpretation of Cohen's *d* values, show about a 10–20% advantage in students' learning. Further, comparison of the two SEMS also shows that the Freshwater baMEL had about a 10% stronger linkage between plausibility and knowledge at post instruction, indicating a greater potential for plausibility reappraisal (i.e., based on a practical interpretation of comparison of Cohen's $f^2$ values).

## 5. Discussion

The current study revealed that both types of MEL diagrams (i.e., the wetlands resources pcMEL and freshwater availability baMEL) promoted students' evaluations, plausibility reappraisals, and knowledge construction about water resources; however, there were differences between the two MEL types. For instance, the baMEL promoted higher levels of evaluation and greater knowledge changes compared to the pcMEL. Based on the results of the SEM, we speculate that higher levels of evaluation led to more pronounced plausibility shifts toward the scientific, which in turn directly led to greater levels of post-instructional knowledge scores. This conforms to Lombardi and colleagues' theoretical framework [14]. One explanation for the differences within the two MELs could be that providing more autonomy in selecting the models and evidence statements motivated children to learn more about the topic [42,43]. Researchers can directly test this hypothesis. A positive result would strengthen and expand our theoretical framework to say that activities like baMEL lead to better plausibility judgments and knowledge gains. In addition, although this study did not examine long-term knowledge changes (i.e., delayed post instructional change), we have some confidence that the gains made by students in the present study would have been sustained based on the results of previous research. For example, Lombardi, Sinatra, and Nussbaum [9] identified that knowledge gains were sustained 6 months post-instruction for students involved in the MEL activities.

Additionally, in the freshwater baMEL, there was a main effect for plausibility shifts toward the scientific and engineering explanations, indicating judgments shifting towards a more scientific and technological stance. These results support our hypothesis that the additional step of students' selection of lines of evidence and explanatory models results in greater conceptual agency, more critical evaluations, and deeper post-instructional knowledge. Additionally, these results indicated that the relations between evaluation, plausibility, and knowledge are more beneficial than evaluation to knowledge directly. This suggests that students who made more scientific evaluations judged scientific and technological models as more truthful than non-scientific alternatives. In turn, more scientific plausibility judgments related to deeper understanding of concepts regarding water resources. Socioscientific issues, including those involving Earth and/or environmental science topics, are subject to differences in the explanations that scientists and the public (students) find plausible [13]. This plausibility gap may be addressed in science classrooms by engaging students in activities that forces them to consider their own judgment of what they know and what scientists know. Results from this study, as well as those from previous studies, suggest an instructional treatment that can help students think more scientifically, by engaging in critique and argument, analyzing lines of evidence, and using models [11,26]. Engaging in such evaluative activities is important because they can potentially promote other scientific practices, such as making observations, collecting data, and forming a research question and hypotheses.

### Limitations

This present study, and other studies that use MEL instructional activities, have provided empirical support about the relations between scientific evaluation, plausibility judgments, and knowledge construction. With the present study being part of a multi-year project, we continue to develop different types of MELs for different Earth and environmental science topics, and in doing so, gain additional

insight into the design, development, and testing of instruction that promotes scientific thinking and deeper knowledge. For example, the present study suggests that the baMEL promotes higher levels of evaluation compared to the pcMEL and previous studies have suggested that the pcMEL is more effective than the MonoMEL [9,26]. With that being said, the nature of this being a classroom-based study (as are all MEL studies) pose some own limitations.

First, although our student samples represent diverse populations, in terms of geographic locations and demographic makeup, we encourage future work in other settings to provide evidence of broad-based applicability of these instructional activities. A key component of doing this is that individuals might have very different experiences and prior knowledge of various Earth and environmental science topics that may influence their learning. A key example is fracking, where individuals living in different areas may have different levels of knowledge and values about this process. These two water topics may be similar in this regard. Although these water resources issues may have not been at the forefront of environmental and human rights issues like climate change, there may be different experiences of these topics in other regions. Based on pre-instructional scores, the participant sample involved in the present study did not seem to have much knowledge of water resources.

One of the most challenging tasks that educators have is trying to keep their students cognitively, behaviorally, and emotionally engaged in their school work [43]. Within science topics, developing students' critical and scientific thinking skills can go hand-in-hand with these three engagement components. With that being said, our previous research has supported the notion that the connection is not that simple. In the classroom, being critical involves understanding the validity of a statement or an explanation based on the ability to interpret the credibility of evidence and the plausibility of explanations. In a day and age where individuals are able to put out false information to millions of people with ease (especially information about controversial socioscientific issues like climate change), it is more important than ever for students to correctly evaluate information [44].

## 6. Implications

In order to address epistemic and social learning goals in science classrooms [6], we have proposed that the practice of explicit and purposeful evaluation needs to be put into place in science curricula, which can, in turn, narrow the plausibility gap (i.e., the difference between what explanations scientists and the public find plausible). Adults may also need to participate in plausibility reappraisal and evaluation of scientific claims in today's world, where misinformation may be distributed through online sources and multiple claims about individual phenomena abound [43]. One solution may be to implement an instructional scaffold like the MELs in university classrooms. However, more research is also needed to properly assess whether the evaluative processes described in this study engage individuals in processes of not only evaluation but also of detecting online misinformation.

Scientists can also alleviate the plausibility gap. The process of evaluation, we reiterate, is perpetual and foundational in science [8]. In a journal article, this may be most evident in the "Discussion" section, in which hypotheses and explanations might be compared. Scientists may even strengthen their comparisons and proposals of explanations by, for example, forming better questions and hypotheses, creating better and more valid measures, and addressing the various contexts in which a phenomenon might arise. Scientists also need to evaluate theories by which they can then explain their data: it is possible that data might fit one theory more than another. Doctoral and Masters programs may therefore be designed so that students are trained to think more scientifically.

Water and wetlands are important topics because of their tenuous nature and because people's misunderstanding of the scope and complexity of the issues within each may render scientific claims implausible [44]; therefore, this general issue warrants attention in the classroom. The two MELs in the present study are part of a larger suite of MEL activities that concern sustainability broadly (e.g., causes of current climate change, impact of hydraulic fracking) and may be used in comprehensive science instruction supporting strategic sustainable development [45–47]. Further, integration of socio-scientific

issues around sustainability topics have the potential to increase the relevance of science instruction so that students become actionable agents for constructive solving in their communities [48,49].

In and of themselves, the MEL activities are not a panacea. The activities are relatively short, with each taking about 90 min of classroom instructional time. As individual lessons, we have designed the MEL activities to fit within a multi-week instructional unit about sustainability. In planning such a unit, the MEL activities could be included with other research-based materials and techniques that facilitate scientific critique and evaluation, and deepen students' science learning. For example, incorporating vignettes within water investigations that identify epistemic considerations involved in scientific modeling of the water cycle could be a complementary lesson within a sustainability unit [50]. Including instructional activities steeped in model-based reasoning could also help support students how to systematically trace water through multiple and branching environmental systems and allow them to construct scientifically accurate explanations of water movement from atomic-molecular to landscape scales [51]. Entire units could be constructed around socio-scientific questions that are locally, regionally, and globally relevant to students, such as equitable and sustainable allocation of water resources [52]. An instructional unit might incorporate scaffolded decision-making steps that facilitate students problem-solving and reasoning to enhance their civic and scientific understanding and increase their agency to become productive and beneficial community members [52]. In such a unit, the MEL activities might be included as one of many lessons designed to engage students in reasoning about scientific evidence and explanations.

## 7. Conclusions

The results of the present study revealed a slight advantage of the baMEL—compared to the pcMEL—in promoting: (a) deeper levels of evaluation between lines of evidence and alternative explanatory models; (b) plausibility shifts toward the scientific alternative; and (c) increased understanding of water resources. Such evaluative practices, which underpin many of the scientific practices found in reformed science instruction, have the potential to deepen students' scientific literacy upon high school graduation [7,8,53]. This study shows that activities facilitating more scientific evaluations and learning are effective instructional tools that educators may employ to deepen their students' conceptual agency to effectively solve local, regional, and global water challenges.

**Author Contributions:** The first two authors (J.M. and J.J.) contributed equally to this research and effectively served as co-first authors in this work, with author order determined by coin-flip. Conceptualization, J.M. and J.J.; Funding acquisition, D.L.; Methodology, J.J.; Project administration, D.L., M.A.H. and C.R.; Supervision, D.L.; Visualization, J.M.; Writing-original draft, J.M. and J.J.; Writing-review & editing, J.M., J.J., D.L., M.A.H. and C.R. All authors have read and agreed to the published version of the manuscript.

**Funding:** This research was funded, in part, by the US National Science Foundation (Grant Nos. 1721041 and 2027376).

**Acknowledgments:** Special thanks to Timothy G. Klavon for contributing to the conceptualization in how to calculate plausibility scores.

**Conflicts of Interest:** The authors declare no conflict of interest. The funders had no role in the design of the study; in the collection, analyses, or interpretation of data; in the writing of the manuscript, or in the decision to publish the results.

## Appendix A

**Table A1.** Racial and ethnic compositions of each school.

| Teacher | *n* | Sex, Race, and Economic Composition |
|---|---|---|
| Teacher 01 | 11 | Female 50.3%, White 81.2%, Hispanic 11.1%, Black 2.8%, Asian 2.8%, Native Hawaiian (pacific islander) 0.2%. Two or more races 2.0%, Economically disadvantaged students 19.3% |
| Teacher 05 | 42 | Female 47.3% White 29.3% Hispanic 43.6%, Black 9.2%, Asian 16.3%, Native Hawaiian 0.1%, Two or more races 1.4%, Economically disadvantaged students, 37.4% (1) |
| Teacher 06 | 16 | Female 44.4%, White 77.1%, Hispanic 7.1%, Black 7.6%, Asian 3.0%, Native Hawaiian 0.3%, Two or more races 4.9%, Economically disadvantaged students 13.3% (1) |
| Teacher 08 | 39 | Female, 49.3% White 44%, Hispanic 20%, Black 8%, Asian 24%, Native Hawaiian not reported, two or more races 3%. Economically disadvantaged students 6% |

## Appendix B

**Table A2.** Models for both pcMEL and baMEL.

| Model | Wetlands pcMEL | Freshwater baMEL |
|---|---|---|
| Non-Scientific (A) | Wetlands are a nuisance to humans and provide little overall environmental benefits. | Earth's freshwater is abundant and will remain so even in the face of global climate. |
| Engineering (B) | - | Earth has a shortage of freshwater that can be met by engineering solutions. |
| Scientific (C) | Wetlands provide ecosystem services that contribute to human welfare and help sustain the biosphere, | Earth has a shortage of freshwater, which will worsen as our world's population increases. |

## Appendix C

Wetlands pcMEL Evidence Statements

1. Wetlands play a role in the global cycles of carbon, nitrogen, and sulfur. Wetland changes these nutrients into different forms necessary to continue their global cycle.
2. Flooding is a natural occurrence in low-lying areas and wetlands are places where floodwaters can collect.
3. Wetlands contribute 70 percent of global atmospheric methane from natural sources.
4. Many wetlands are located in rapidly developing areas of the country.

Freshwater baMEL Evidence Statements

1. Land use changes have generated large pressures on freshwater resources. These changes are affecting both water quality and availability.
2. The world's population is increasing. This stresses the supply of freshwater.
3. Groundwater provides freshwater to many people around the world. In many places, people are using groundwater faster than it is replaced by precipitation.
4. Water reclamation costs have gone down in the past several years. These costs vary depending on location. Making sea water drinkable costs more than reclamation.
5. Advances in engineering have led to better access to quality drinking water. At the same time life expectancy and quality of life have improved.
6. Glaciers are a source of freshwater in many parts of the world. Glacial ice mass is decreasing worldwide.

7. Most climate predictions are on regional scales. Microclimates are local areas where precipitation and temperature are influenced by vegetation cover, topography, and human activity. Large-scale predictions may not accurately reflect local trends in freshwater availability

8. In the contiguous US, average temperatures and precipitation have increased since 1901. From 2000–2015, the US was abnormally dry with some parts of the country in moderate to severe drought.

## Appendix D

**Table A3.** Explanation task rubric.

| Category | Description | pcMEL Example | baMEL Example |
|---|---|---|---|
| Erroneous Evaluation | Explanation contains incorrect relationships between evidence and model, excluding misinterpreting a "Nothing to Do With" relationship by elimination-based logic. The explanation may also be mostly inconsistent with scientific understanding and/or include nonsensical statements. | "It shows the negative effects." | "Groundwater is abundant enough where water depletion won't have an effect." |
| Descriptive Evaluation | Explanation contains a correct relationship without elaboration, or correctly interprets evidence without stating a relationship. For example, the evidence-to-model link weight states that the evidence has nothing to do with the model. Explanation does not clearly distinguish between lines of evidence and explanatory models. Explanations could also demonstrate "elimination-based logic" to come to a positive or negative weight, when evidence-to-model link weight states that the evidence has nothing to do with the model. For example, an explanation states that an evidence supports one model, but uses reasoning that the evidence contradicts the other model. | "It explains all the benefits wetlands gives to humans and the environment" | "As the population increases, more resources are required" |
| Relational Evaluation | The explanation addresses text similarities, and includes both specific evidence and an associated model or reference to a model. For example, explanation is correct, with an evidence-to-model link weight of strongly supports, supports, or contradicts as appropriate. Explanation distinguishes between lines of evidence and explanatory models, but does so in a merely associative or correlation manner that is often based on text similarity. | "This shows the benefits of wetlands are throughout the biosphere, so destroying them would have broad implications." | "It states that people are using groundwater faster than its replacement, which means as population increases, it will make the problem of availability worse." |
| Critical Evaluation | Explanation describes a causal relationship and/or meaning of a specific relationship between evidence and model. For example, explanation is correct, with an evidence-to-model link weight of strongly supports, supports, or contradicts as appropriate and reflects deeper cognitive processing that elaborates on an evaluation of evidence and model. Explanation distinguishes between lines of evidence and explanatory models, allows for more sophisticated connections, and/or concurrently examines alternative models. | "Flood is a big problem and poses a threat to human safety. Wetlands are able to hold in the flood water, so the destruction of wetlands reduces possibilities of flooding for cities and towns." | "Evidence #2 says that in the next 30 years, there will be a change from 7.5 to 9.5 billion people on this earth, With every person requiring 50 L of freshwater per day already causing a stress on the current water situation, the increase in future population with cause an even bigger stress and supports model C that a shortage will worsen." |

## Appendix E

Students rated the degree to which hydrologists/scientists agree with the statements. Scores were coded from 1 through 5 (1 = Strongly disagree, 5 = Strongly agree). Items marked with * are reverse-coded.

Wetlands pcMEL Knowledge Items

1. Wetlands occur naturally on every continent.
2. Loss of wetlands will have little impact on human welfare.
3. Frogs need wetland habitats in which to reproduce and feed.
4. Draining of some wetlands can result in release of carbon to the atmosphere, which could increase global warming.
5. Wetlands cause sudden and damaging floods downstream.

Freshwater baMEL Knowledge Items

1. Water reclamation makes contaminated water safe for humans to use.
2. Engineers will solve current shortages of freshwater.
3. Freshwater is abundant and will remain so even in the face of global climate change.
4. Land use decisions affect Earth's surface, but have little impact on the water cycle.
5. Technology advances have made water safer for human use.
6. Groundwater recharge rates are similar from place to place because soils are generally uniform.
7. Global temperatures have increased. But, there has not been an overall decrease in global glacial ice.
8. Microclimates have various levels of precipitation. This affects how much water is available for human use.
9. Over the past 100 years, lower amounts of rainfall have occurred across the US. This means that greater amounts of land have been affected by drought in the last 20 years.
10. Current shortages of freshwater will get worse around the globe as world population increases.
11. Climate change and increasing populations will lead to more freshwater shortages.
12. Depletion of groundwater causes land to sink. Depletion also causes freshwater to be contaminated.

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
