# Peer review of "Students’ Scientific Evaluations of Water Resources"

_water, doi:10.3390/w12072048_

Round 1

Reviewer 1 Report

Presented article deals with very important socio-scientific issue. It is quite well structured from scientific point of veiw - but it needs more attention from pedagogical aspects. Few places needs clarification - like the whole paragraph from line 191 to 202. More detailed concerns are listed below:

1) in the theoretical framework especially part dedicated to evaluation (part 2.1) there is a gap in the role of evaluation in educational processes - which is crucial in presented work. In my opinion the article would benefit from providing some arguments for it as well not only the role of evaluation in science. We just might find closing sentence dedicated to it: "in order to promote evaluative processes is to explicitly engage students in judgments about knowledge and knowing (i.e., epistemic judgments, such as judging the plausibility of scientific explanations)." but this one does not have support in the literature source - so it is needed here.

2) if the authors wanted to examine: "the relationship between engagement in evaluating" it should be a section in the theoretical background dedicated to it as well, it is important part of learning - one part is cognition and cognitive processes but the second deals with emotions and motivations, so engagement  - as a factor investigated by authors should be explained - how do they understand it, which perspective they pick for their research. Or if it wasn't the factor (and in my opinion it wasn't really) that the authors should reformulate the research questions.

it is also connected to the first research question - what kind of levels of evaluation did the authors expect? This is later in text somehow explained, but it should be obvious for the reader from the beginning. Because in this place it appears as something new and surprising - and this should not be the case (at least in my opinion).

3) It is not clear for me from the text and I am also wondering whether students are familiar with the idea of plausibility? Did they have on the previous lessons (or in the curriculum) dedicated to how to distinguished scientific from non-scientific information? What are the important factors they should be looking for why making judgements about plausibility?

4) the procedure seems to be very complicated – the reader has to be very focused on each step. Maybe it would be good idea to provide some graphical representation of the procedure?

5) some editorial work needs to be done as well  - eg.line 349 – the word instructional is repeated twice, or 396 – 397 “there  was main effect was significant for time,”, Table XX.

6) every single graph and chart in the article should be self-explanatory, in this case I am a bit lost, for example fig 3. Evaluation scores (range: 1-4) – it should be written what 1 means what 4, because in nominal scale it looks like the high school students performed worse than middle school. Especially that the authors are using many different ways of measuring things – and a reader has to go back and forth to search for the scale> And this remark is valid for every table

7) I wonder how would the authors explain why post tests were done just immediately after the intervention and not any more? Usually it is not surprising to receive better results in anything just after the intervention. Why have you decided to do not check what stays in students’ mind after let’s say 3 months? (with such usage of statistical analysis not providing postponed test?)

8) the authors claim: These results support our hypothesis… but they did not provide any hypothesis, there are just research questions really explicit.

9) the way authors presents results section – mostly statistical analysis - is difficult to follow. Again I would recommend some graphical representations and more reminding parts  - for what each abbreviation stands for (eg, A, B and C).

10) discussion is very narrow, there is a massive literature on the topic not used here at all.

To sum up

The beginning is very nice, but later on the article loses its clarity, (also from editorial perspective) and its flow (the results and discussion section needs much more attention). Article would benefit from graphical organizers and representations  to help reader to follow the major ideas.

Reviewer 2 Report

The authors used ANOVA and a structural equation model to understand how environmental science students can deepen their knowledge about water resources through an instructional scaffold. In general, this paper is well-written, but it can be improved in the following fronts.

First, as this paper seems to build largely on Lombardi and his colleagues’ work, the authors should clarify what derived knowledge/findings/insights are new to this field. In particular, the theoretical contributions should be highlighted.

Second, the paper has the potential to engage more with the wider literature in the scientific understanding of sustainability to attract more audience. For example, can the instructional activities at different levels contribute to a more comprehensive understanding of sustainability? Suggest reading, for instance, https://www.sciencedirect.com/science/article/pii/S0959652615015930?casa_token=fwOJ85uo1Z0AAAAA:vb6uVYRmsPbwDgHmDEqIwg2zYGF_gexRNE4buvCLAw80siwz0vaFgEfjsDIyPCmPiBMo5p8d; https://www.mdpi.com/2071-1050/11/16/4281;  and  https://link.springer.com/article/10.1007/s10668-006-9058-z

Third, the conclusion section discusses too much about future steps rather than focusing on this particular study. Clearer and more specific connections between the introduction and conclusion sections are expected.

Despite the weaknesses mentioned above, this work is methodologically sound, timely and interesting. I therefore suggest a minor revision.

Round 2

Reviewer 1 Report

The authors have improved their manuscript in large part. Now it is more clear - my congratulations. But the discussion is still very superficial and shows huge gap - the manuscript would benefit from presenting the results in the lights of more scientific content. Not only Lombardi is writing about it :-) This is my only remark (but quite important) to the manuscript.

Author Response

Authors' response: We have added a paragraph in the discussion (lines 630-646) to include more research in this area. We hope this satisfies the reviewer's concerns of superficiality and for us to be more inclusive of the high-quality research that is being conducted around the context of water resource and sustainability instruction. However, if the reviewer is still concerned, we want to emphasize that it is not our intent to imply that Lombardi and colleagues are the only ones doing work in this area, and we would welcome specific suggestions for citations to include in the discussion, with the ultimate hope of being productive members of this research community.